# The Cytoplasmic Region of SARAF Reduces Triple-Negative Breast Cancer Metastasis through the Regulation of Store-Operated Calcium Entry

**DOI:** 10.3390/ijms24065306

**Published:** 2023-03-10

**Authors:** María Paz Saldías, Pablo Cruz, Ian Silva, Octavio Orellana-Serradell, Boris Lavanderos, Diego Maureira, Raquel Pinto, Oscar Cerda

**Affiliations:** 1Millennium Nucleus of Ion Channel-Associated Diseases (MiNICAD), Santiago 8380453, Chile; 2Program of Cellular and Molecular Biology, Institute of Biomedical Sciences (ICBM), Faculty of Medicine, Universidad de Chile, Santiago 8380453, Chile

**Keywords:** triple-negative breast cancer, store-operated calcium entry, SARAF, cell migration, cell invasion, metastasis

## Abstract

Triple-negative breast cancer has a poor prognosis and is non-responsive to first-line therapies; hence, new therapeutic strategies are needed. Enhanced store-operated Ca^2+^ entry (SOCE) has been widely described as a contributing factor to tumorigenic behavior in several tumor types, particularly in breast cancer cells. SOCE-associated regulatory factor (SARAF) acts as an inhibitor of the SOCE response and, therefore, can be a potential antitumor factor. Herein, we generated a C-terminal SARAF fragment to evaluate the effect of overexpression of this peptide on the malignancy of triple-negative breast cancer cell lines. Using both in vitro and in vivo approaches, we showed that overexpression of the C-terminal SARAF fragment reduced proliferation, cell migration, and the invasion of murine and human breast cancer cells by decreasing the SOCE response. Our data suggest that regulating the activity of the SOCE response via SARAF activity might constitute the basis for further alternative therapeutic strategies for triple-negative breast cancer.

## 1. Introduction

Breast cancer is the most common type of cancer and the leading cause of death in women worldwide [1,2]. It is characterized as a heterogeneous disease at the molecular level, and both prognosis and treatment depend on a subclassification determined by the histological and molecular features of the tumor [3]. The triple-negative breast cancer (TNBC) subtype represents approximately 15% of the total breast cancers and is characterized by an aggressive subtype that has a 40% recurrence rate owing to metastatic spread [4]. Since TNBC tumors lack estrogen and progesterone receptors, and HER2 expression, they are not susceptible to first-line therapies, such as hormone-based therapy. Hence, there is an urgent need to develop new therapeutic strategies to overcome the poor prognosis of this cancer subtype.

Ca^2+^ signaling is deregulated in several types of cancers [5,6]. In different types of cancers including breast cancer, changes in the expression and activity of entities, such as ion channels, transporters, and their regulators [7,8] have been proposed as tumorigenic events that lead to malignancy. Altered Ca^2+^ homeostasis increases cell motility and invasion, proliferation, apoptosis resistance, and changes in gene transcription (as reviewed by Monteith et al. (2017) [6]). The store-operated Ca^2+^ entry (SOCE) response is a ubiquitous and essential cellular homeostatic mechanism that regulates Ca^2+^ levels in both excitable and non-excitable cells. This response is mainly mediated by Orai1, a highly selective Ca^2+^ channel located in the plasma membrane (PM). Its activity relies on its interaction with STIM1, a type I transmembrane protein located mainly in the endoplasmic reticulum (ER). In the absence of Ca^2+^ in the ER lumen, STIM1 leads a clustering process that allows the formation of the Orai1-STIM1 complex at the ER-PM junctions, activating Orai1 channels. The SOCE response is relevant in different types of cancer such as melanoma [9,10,11], glioblastoma [12], clear cell renal carcinoma [13], hepatocellular carcinoma [14], prostate cancer [15], cervical cancer [16] and breast cancer [17,18,19,20]. These types of cancer exhibit an augmented SOCE response and overexpression of STIM1 and/or Orai1. The inhibition of SOCE decreases the proliferation and metastasis of cancer cells [21]. Similarly, increased Orai1-STIM1 activity enhances the formation of podosomes and invadopodia, thus contributing to the invasive phenotype of breast cancer cells [22,23]. These reports suggest a robust and novel role of Orai1-STIM1 activity, characterized by an increased expression of Orai1-STIM1 which contributes to cancer progression and metastasis. Thus, understanding the molecular mechanisms that regulate Orai1-STIM1 activity could help in the development of new therapeutic strategies to treat TNBC, particularly by taking advantage of the upregulated SOCE response.

Only a few molecules are known to regulate the activity of the Orai1-STIM1 complex [24,25,26,27,28,29]. The SOCE-associated regulatory factor (SARAF), a single-pass type I ER membrane protein, is the most effective regulator of Orai1-STIM1 channels [30]. Recently, Zomot et al. (2021) described a dual role for SARAF in modulating CRAC channel activity as well [31]. Upon ER-Ca^2+^ depletion, SARAF facilitates the release of the SOAR domain of STIM1 by promoting its activation and translocation to the ER-PM junctions, and the subsequent anchoring of the Orai1 channel at those microdomains [31]. During SOCE response activation, post-ER Ca^2+^ depletion, SARAF-STIM1 complexes dissociate, allowing STIM1 oligomerization [30,32]. The SARAF-STIM1 complex reassociates and mediates the slow Ca^2+^-dependent inactivation (SCDI) of the Orai1 channel, thus preventing Ca^2+^ overload [30,32]. The heterologous expression of SARAF decreases the Ca^2+^ release-activated Ca^2+^ current (*I*_CRAC_) and SOCE [30,32]. These effects depend on the binding of the cytoplasmic C-terminal region of SARAF to the C-terminal inhibitory domain (CTID) of STIM1 [30,32]. 

As such, the SARAF-STIM1 interaction constitutes the main inhibitory molecular checkpoint for SOCE regulation. It is proposed that SARAF plays a crucial role in the development and progression of breast cancer, although no significant differences have been observed in SARAF protein levels between pre-neoplastic and neoplastic breast cancer cell lines [33]. The silencing of the SARAF protein improves SOCE component expression in pre-neoplastic and TNBC cells. At the same time, in the luminal subtypes, the overexpression of SARAF decreases the expression of SOCE components. Furthermore, SARAF silencing reduced proliferation and migration in MCF7 and MDA-MB-231 cells, whereas MCF10A cells were unaffected, suggesting its cell-type-dependent differential role [33]. Therefore, the identification of novel modulators of the SARAF-STIM1 interaction and their impact on Orai1-STIM1 function is an interesting biomedical research topic. Currently, the lack of drug specificity remains a challenge, despite the attempts to find selective pharmacological modulators of Orai1-STIM1 in different models [34]. It is known that the cytoplasmic C-terminal region of SARAF protein is sufficient to exert its activity as negative regulator of the SOCE response [30]. Here, we determined the effect of the cytoplasmic C-terminal region of SARAF on the SOCE response in breast cancer and in the critical cellular events leading to the malignant phenotype, such as tumor cell proliferation, motility, and metastasis [35]. Our data suggest that regulating the activity of the SOCE response via SARAF activity might constitute the basis for further alternative therapeutic strategies for triple-negative breast cancer.

## 2. Results

### 2.1. SARAF Expression Is Reduced in TNBC

To evaluate whether canonical SOCE components including the Orai1, STIM1, and SARAF proteins, are correlated with breast cancer metastatic potential, we consulted The Cancer Genome Atlas Program (TCGA) NIH database using the GEPIA platform [36]. Results indicate that no statistically significant changes in Orai1, STIM1, and SARAF expression were observed in tumorous and non-tumorous tissues (Figure 1A–C). SARAF expression is diminished in the triple-negative subtype compared to the Luminal A subtype of tumors (Figure 1C). The Luminal A subtype manifests less metastatic potential and thus presents a better prognosis than TNBC. In addition, we evaluated the endogenous protein levels of the SOCE components in different breast cancer cell lines (Figure 1D,E). Relative quantification indicated that the expression of the components depended on the aggressiveness of the cell line (Figure 1E). As the MCF10A line is a non-neoplastic cell line, we observed a decrease in STIM1 expression in MCF7 and MDA-MB-231 cells. No differences were observed in Orai1 expression in MDA-MB231 cells. However, we observed a diminished expression of the Orai1 protein in T47D and MCF7. Notably, SARAF expression diminished in all malignant cell lines. Therefore, since triple-negative tumors have high metastatic potential compared to other tumors, decreased SARAF expression can be correlated to a higher migration and invasive tumor phenotype. It also suggests that the overexpression of SARAF protein could be a potential strategy to counteract the highly invasive and proliferative behavior of TNBC cell lines.

### 2.2. The C-Terminal SARAF Fragments Reduce Store-Operated Calcium Entry

It has previously been reported that the C-terminal region of SARAF is sufficient to decrease the SOCE response [30]. As a decrease in SOCE response affects malignancy of breast cancer cells, we generated a C-terminal fragment of SARAF fused to mScarlet fluorescent protein (mScarlet-C-SARAF) (Figure 2A) and evaluated the effect of its overexpression on the SOCE response. Our initial approach was to determine the SOCE response via Ca^2+^ imaging, using Fura-2-AM in HEK293 cells. We observed that the overexpression of mScarlet-C-SARAF decreased the SOCE response without affecting thapsigargin-induced Ca^2+^ release (Figure 2B–D). Moreover, we found that mScarlet-C-SARAF interacts with STIM1 (Figure 2E). Once the effect of this construct was validated in HEK293 cells, we performed Ca^2+^ imaging by overexpressing mScarlet or mScarlet-C-SARAF in the human TNBC cell line MDA-MB-231. We observed that the overexpression of mScarlet-C-SARAF resulted in similar results to those obtained in HEK293 cells (Figure 2F–I). We also evaluated the effect of the overexpression of the mScarlet-C-SARAF fragment on the formation of the STIM1-Orai1 complex in MDA-MB-231 cells. For this purpose, we performed a bimolecular fluorescence complementation (BiFC) approach [37,38], using STIM1 and Orai1 constructs, which are fused to either non-fluorescent N or C terminal fragments of the Venus protein [39] (Figure 2J,K). We observed that mScarlet-C-SARAF overexpression decreased puncta formation upon thapsigargin stimulation, indicating that the interaction between STIM1 and Orai1 is reduced. This also correlates with a diminished nuclear translocation of NFAT, indicating that a diminished SOCE response elicited by mScarlet-C-SARAF expression affects the signaling associated with this response (Figure 2L,M). These results together indicate that mScarlet-C-SARAF overexpression reduces the SOCE response.

### 2.3. The C-Terminal SARAF Fragment Reduces Malignancy Features in Human TNBC

The SOCE response regulates proliferation, apoptosis resistance, migration, and invasion in different types of cancer [21]. We evaluated the effect of C-SARAF overexpression in MDA-MB-231 cells using MTT cell proliferation assays. The results indicate that the overexpression of the mScarlet-C-SARAF fragment had no effect on cell proliferation compared to the control between 16 and 48 h (Figure 3A). Scratch assay results indicate (Figure 3B) a decrease in cell migration when overexpressing the mScarlet-C-SARAF fragment compared to the control condition (mScarlet overexpression) (Figure 3B,C). We also performed Transwell^®^ invasion assays in MDA-MB-231 cells to determine if the overexpression of the mScarlet-C-SARAF fragment was sufficient to reduce cell invasion. We observed that mScarlet-C-SARAF-overexpression reduced cell invasion, confirming that SOCE inhibition by the C-terminal fragment of SARAF could reduce the metastatic potential of TNBC cells (Figure 3D,E). The above results suggest that the mScarlet-C-SARAF fragment regulates the SOCE response, reduces Ca^2+^ entry, and modulates the migratory and invasive potential of TNBC cell lines, but not their proliferation.

### 2.4. The C-Terminal SARAF Fragment Affects Dynamic Focal Adhesion and Cell Retraction

Since mScarlet-C-SARAF reduces the migratory and invasive potential of the human TNBC cell line, MDA-MB-231, we evaluated if the overexpression of cytoplasmic C-terminal SARAF could affect cellular processes, such as focal adhesion (FA) dynamics and cytoskeleton rearrangement, both involved in cell migration and invasion. To explore these processes and to determine the adhesion capacity of cells overexpressing C-SARAF, we performed cell spreading assays. We observed that mScarlet-C-SARAF reduced the area of MDA-MB-231 cells after 45 min of seeding (Figure 3F–H).

FA dynamics is a cellular process involving coordination between FA turnover and actin cytoskeleton rearrangement, and it is the key for cell migration [40]. To determine if C-SARAF affects focal adhesion turnover, we co-transfected MDA-MB-231 cells with EGFP-paxillin and mScarlet or mScarlet-C-SARAF and analyzed the FA dynamics using TIRF microscopy (Figure 4A). The results suggest that the mScarlet-C-SARAF fragment increased the assembly rate (Figure 4B) without affecting the disassembly rate (Figure 4B–E). In addition, we observed a decrease in cell retraction (Figure 4E,F), indicating that the overexpression of C-SARAF impacts focal adhesion dynamics and cell motility. A decrease in cell movement is likely due to a decrease in cell retraction upon serum stimulation. In summary, mScarlet-C-SARAF overexpression increased the assembly rate of FA and reduced cell retraction. 

### 2.5. The C-Terminal SARAF Fragment Reduces SOCE Response, Malignancy Features and Metastasis, and Promotes Lymphocytic Infiltrate in the Primary Tumor TNBC in a Murine Model

To evaluate if the overexpression of mScarlet-C-SARAF in the human TNBC cell line MDA-MB-231 has the same effects in a murine model, we performed proliferation, migration, and invasion assays in the triple-negative murine cell line, 4T1. The results suggest that overexpression of mScarlet-C-SARAF resembled the effects of C-SARAF overexpression in MDA-MB-231 cells (Figure 3), showing a reduced SOCE response, diminished translocation of NFAT from the cytoplasm to the nucleus, cell migration, and invasion in the 4T1 murine cell line (Figure 5A–H). To assess the effect of mScarlet-C-SARAF overexpression in a pathophysiological context, we used a murine model of tumor growth and metastasis. We used an in vivo syngeneic murine model of tumor growth. 4T1 cells (BALB/c strain) stably transfected with mScarlet or mScarlet-C-SARAF were injected into the fourth inguinal mammary gland to generate the primary tumor (Figure 5N). No significant difference was observed in primary tumor size between both conditions (Figure 5N,O). However, mice injected with cells expressing the mScarlet-C-SARAF fragment had a lower percentage of liver vessels affected by infiltrating tumor cells compared to the control (Figure 5P,Q). 

## 3. Discussion

Several studies have shown that in breast cancer, the regulation of Ca^2+^ signaling is crucial for tumorigenesis and cellular processes, such as cell growth, proliferation, migration, metastasis, and resistance to apoptosis (reviewed in [5,41]). In Ca^2+^ signaling, remodeling of the composition and activity of the protein complexes involved is a common phenomenon. Proteins involved in the SOCE response, such as Orai [20,42], TRPC1 [43,44], TRPC6 [45,46] and STIM [20,47,48,49] act as modulators of the Ca^2+^ homeostatic response. In fact, evidence indicates that the increased SOCE response is key for tumorigenic processes, such as proliferation, metastasis, tumor formation, and the avoidance of antitumor immunity [50]. Nevertheless, although only few regulators of the SOCE response are involved in this process, the development of compounds targeting SOCE has been studied as a potential strategy for cancer therapy, including breast cancer [51]. As these compounds are specific and exhibit side effects, the study of the interactors or regulators of the SOCE response has become an interesting pharmacological target. Herein, we propose the role of the C-terminal region of SARAF as a potential negative regulator of the SOCE response in TNBC. 

When SARAF was first identified, Palty et al. (2012) reported that overexpression of the C-terminal region of SARAF reduced the SOCE response efficiently [30] and in our research, it reduced Ca^2+^ entry during Fura-2-AM recordings in HEK293 (Figure 2A–D), MDA-MB-231 (Figure 2F–I) and 4T1 cells (Figure 5A–C). Furthermore, the mScarlet-C-SARAF fragment induced a reduction in the SOCE response and led to a reduced STIM1-Orai1 complex formation, as evidenced by the BiFC assay in a human TNBC cell line. These data indicate that a reduction in the SOCE response in breast cancer cells could be related to the inhibition of STIM1-Orai1 complex assembly, as reported previously [30].

Different reports have shown the relevance of the SOCE response and its effect on cell migration and the invasion of neoplastic cells [20,52,53]. A reduction in the expression of STIM1 and Orai1 diminishes cell migration and invasion, and decreases metastasis in a murine model [12,20]. Our results suggest that the mScarlet-C-SARAF fragment reduced the SOCE response and resulted in lower migration and the invasion of MDA-MB-231 and 4T1 cells without affecting cell proliferation. This effect can be explained by an alteration in the regulation of focal adhesion formation.

Currently, the molecular mechanisms involved in the SOCE-dependent regulation of focal adhesions need further investigation. D’Souza et al. (2020) showed that the Orai1-STIM1 complex clusters near the FAs of MDA-MB-231 cells and Ca^2+^ influx via SOCE leads to activation of the GTPase ARF5 via the Ca^2+^-activated GEF IQSec1, which is essential for promoting focal adhesion disassembly [54]. In an osteosarcoma model, Lin et al. (2021) showed that constitutively active STIM1 increased Ca^2+^ influx and the turnover of FA proteins, which impeded cell migration [55]. In contrast, dominant-negative STIM1 decreased FA turnover and promoted cell migration. Another protein has been reported to regulate the conformational change in STIM1, promoting its activation in TNBC. Lee et al. (2022) determined that PGRMC1, a protein that is expressed in TNBC, promotes the SOCE response, in addition to regulating the turnover of focal adhesions and the formation of actomyosin [56]. These data suggest that cancer cells require sufficient Ca^2+^ to control the assembly and disassembly of focal adhesions. Our data show an increase in the rate of assembly of focal adhesions and a decrease in cell retraction in the mScarlet-C-SARAF overexpressed condition in MDA-MB-231 cells, consistent therefore with a lower migration potential. Thus, the mScarlet-C-SARAF fragments could interact directly with STIM1, regulating its conformational change, which may result in reduced cell migration and the invasion of cancer cells.

During cell invasion, STIM-Orai1 is essential for orchestrating the activation of metalloproteinases [11,57] and the formation of podosomes [23] that allow the degradation and migration of neoplastic cells through the extracellular matrix. Sun et al. (2014) described that oscillations of Orai1 activity have an impact on invadosome formation and the activation of the metalloproteinase MT1-MPP [11]. In addition, Rizaner et al. (2016) reported the high metastatic potential of MDA-MB-231 cells due to spontaneous Ca^2+^ oscillations, whereas such oscillations were not observed in MCF7 cells, which had a reduced metastatic potential [58]. It has been reported that in MDA-MB-231 cells, the expression of Orai1 predominates over Orai2, and this is associated with the activation of NFATc1 with low concentrations of agonists [59]. Taken together, these data indicate that mScarlet-C-SARAF mainly regulates Orai1, which could influence the activation of MPP and the formation of podosomes, implying diminished cell invasion.

Chakraborty et al. (2016) described the use of phemindole, a synthetic dietary indole that exhibits antitumor properties and is proposed as a potential compound for pharmacological use in chemotherapy. Phemindole treatment reduces STIM1 expression by reducing the interaction between STIM1 and Orai1, resulting in decreased cell migration in vitro through the regulation of FAK; a reduction in tumor growth in vivo was also observed [60].

In summary, our data suggest that mScarlet-C-SARAF fragments are a potential alternative to treat breast cancer subtypes such as TNBC due to their inhibitory activity upon the SOCE response and the subsequent reduction in cell migration and invasion. Thus, methods to selectively deliver mScarlet-C-SARAF fragments to tumors are necessary to prevent cancer cell dissemination.

## 4. Materials and Methods

### 4.1. Cell Culture and Transfections

HEK293 (ATCC, CRL-1573) and MDA-MB-231 (ATCC, HTB-26) cells were cultured in DMEM High Glucose media supplemented with 10% *v*/*v* FBS and 100 μg/mL penicillin-streptomycin. The 4T1 (ATCC, CRL-2539) cell line was cultured in RPMI media supplemented with 10% *v*/*v* FBS. MDA-MB-231 cells were transfected using Lipofectamine LTX reagent (Thermo Fisher Scientific, Waltham, MA, USA, Cat#15338100) and 4T1 cells were transfected with the different plasmids using the Lipofectamine 2000 reagent (Thermo Fisher Scientific, Waltham, MA, USA, Cat#11668019). All cell lines were grown at 37 °C and with 5% CO_2_.

Stable lines of 4T1 cells that overexpress the fluorescent protein mScarlet and the fused proteins mScarlet-C-SARAF were generated using the antibiotic G418 (Gibco, Thermo Fisher Scientific, Waltham, MA, USA, Cat#11811-031) at a concentration of 2 mg/mL. These cells were subsequently sorted at the REDECA facility (Faculty of Medicine, Universidad de Chile) using a FACSAria™ III Cell Sorter by selecting cells that exhibited red fluorescence for use in the in vivo metastasis experiments.

### 4.2. Plasmids, Antibodies and Reagents

The plasmids used are summarized in Table 1.

The mScarlet-C-SARAF plasmid was generated from SARAF-myc-DDK (Origene #RC201864) as a template for the amplification of the C-SARAF sequence (amino acids 195 to 339, according to the Uniprot database). Primers with restriction sites 5′-GAG CTC AAG ACG GGC AGT ATT CTC CTC CAC-3′ and 5′-CTG CAG TTA TCG TCT CCT GGT AC-3′ were designed and the C-terminal SARAF sequence was incorporated into *Sacl* and *Pstl* sites of the pmScarlet-H_C1 plasmid (Addgene #85043).

The antibodies used for immunoblot are listed in Table 2.

### 4.3. Ca^2+^ Imaging

Cells were loaded with 5 μM Fura-2-AM (Thermo Fisher Scientific, Waltham, MA, USA, Cat#F1201) for 30 min at room temperature. Then, cells were washed once and recorded using Ringer’s modified medium at pH 7.4 containing 140 mM NaCl, 2.5 mM KCl, 10 mM Glucose, 2 mM CaCl_2_, 1 mM MgCl_2_ and HEPES 10 mM. ER Ca^2+^ depletion was induced with 2 μM thapsigargin (TG) (Merck, Darmstadt, Germany, Cat # 67526-95-8) in free-Ca^2+^ Ringer’s modified medium containing 1 mM EGTA in the absence of CaCl_2_. Samples were excited using an halogen light source with alternated 340 nm and 380 nm excitation filters (Chroma). Fluorescence emissions at 510 nm were captured using a Chameleon Camera (Chameleon CM3-U3-31S4, FLIR, Richmond, BC, Canada), and an Eclipse Ti2-U inverted microscope (Nikon, Tokio, Japan). The SOCE protocol in 4T1 and MDA-MB-231 consisted of 3 min of basal recording with Ringer’s modified with 2 mM CaCl_2_, 7 min of reticular Ca^2+^ depletion using Ca^2+^-free Ringer’s with 2 µM TG to finally replace the milieu with modified Ringer’s 2 mM CaCl_2_ solution, and recording for 5 min. Fluorescence intensity was quantified using the Image J2/FIJI v.2.9.0 software (National Institute of Mental Health, Bethesda, MD, USA) and data were normalized using the initial ratio of fluorescence 340 nm/380 nm.

### 4.4. Immunoblot Analysis

Cells were lysed in lysis buffer containing: 50 mM Tris-HCl, 150 mM NaCl (Merck, Darmstadt, Germany, Cat#1064045000), 1 mM ethylenediaminetetraacetic acid (EDTA; Chemix, Lampa, Santiago, Chile, Cat#160408), 1 mM sodium orthovanadate (Calbiochem, San Diego, CA, USA, Cat#567540), 5 mM NaF (Sigma-Aldrich, San Louis, MO, USA, Cat #S7920), 1% v/v Triton X-100 (Sigma-Aldrich, San Louis, MO, USA, Cat#10789704001), protease inhibitors 1 mM phenylmethylsulfonyl fluoride (PMSF; Sigma-Aldrich, San Louis, MO, USA, Cat#78830) and a protease inhibitor cocktail (PIC; Cytoskeleton, Inc, Denver, CO, USA, Cat#PIC02), adjusted to pH 7.4 for 30 min at 4 °C. The lysates were centrifuged at 12,000× *g* at 4 °C for 10 min. Reducing Sample Buffer (RSB) [62.5 mM Tris-HCl, 2% *w*/*v* SDS (Sigma-Aldrich, San Louis, MO, USA, Cat#L5750), 10% *v*/*v* glycerol (Merck, Darmstadt, Germany, Cat#356352), 1% *v*/*v* β-Mercaptoethanol (Merck, Darmstadt, Germany, Cat #8057400250)] was added to the supernatant samples, and then boiled for 5 min, followed by size fractionation on SDS–PAGE. Following SDS-PAGE, proteins were transferred to nitrocellulose membranes (GE Healthcare Life Sciences, Chicago, IL, USA, Cat#10600002), which were then blocked for 1 h with BLOTTO [4% w/v nonfat dry milk/ 0.1% *v*/*v* Tween-20 in Tris-buffered saline (TBS: 50 mM Tris-HCl, 150 mM NaCl, adjusted to pH 7.5)] followed by 2 h or overnight incubation with primary antibodies. After 3 washes with BLOTTO, the membranes were incubated with the appropriate HRP-conjugated secondary antibody for 1 h. After 3 washes for 10 min each with 0.1% v/v Tween-20/TBS, immunoblots were visualized using Pierce ECL Western Blotting Substrate (Thermo Fisher Scientific, Waltham, MA, USA, Cat#34080). The images were acquired with a Mini HD9 imager (Uvitec Ltd., Cambridge, UK) and quantified measuring the optical density of the bands with Image J2/FIJI v.2.9.0 software.

### 4.5. Immunoprecipitation

Immunopurification assays were performed as previously described [61]. Briefly, HEK293 transfected with mScarlet and mScarlet-C-SARAF were solubilized in lysis buffer containing 1% *v*/*v* Triton X-100, 150 mM NaCl, 1 mM EDTA, 50 mM Tris-HCl (pH 8.0), 1 mM NaVO_4_, 5 mM NaF, 1 mM EDTA, 1 mM phenylmethylsulfonyl fluoride (PMSF) and protease inhibitor cocktail for 15 min at 4 °C, followed by centrifugation at 11,000× *g* for 10 min at 4 °C. The supernatants were incubated with anti-RFP antibodies (Table 2) for 3 h at 4 °C on rotation, followed by the addition of protein G sepharose beads (Cytiva, Marlborough, MA, USA, Cat #17061802) for 1 h at 4 °C on rotation. The beads were washed six times in lysis buffer and immunopurified complexes were eluted by boiling in Reducing Sample Buffer for 5 min, followed by size fractionation on SDS-PAGE and analysis by immunoblot.

### 4.6. Biomolecular Fluorescence Complementation (BiFC)

BiFC assays were performed as described [62]. Briefly, MDA-MB-231 and 4T1 cells were co-transfected with mScarlet or mScarlet-C-SARAF and pcDNA3.1-STIM1-Venus-173-C (STIM1-VC) and pcDNA3-Venus-173-N-Orai1 (Orai1-VN) plasmids [39]. Forty-eight hours post-transfection, STIM1-Orai1 complementation was stimulated by the addition of 2 µM TG for 45 min at 37 °C in Ringer’s solution modified with 2 mM CaCl_2_. Then, the cells were fixed and permeabilized as described in the previous sections. Images were acquired with 63× objective using optical sectioning and structured illumination Zeiss Apotome.2, Axiovert 7. Particle number per cell area was quantified using the Image J2/FIJI v.2.9.0 software.

### 4.7. NFAT Translocation

MDA-MB-231 and 4T1 cells were co-transfected with mScarlet or mScarlet-C-SARAF and EGFPC1-huNFATc1EE-WT [63] plasmids and then plated on polylysine-coated coverslips. Forty-eight hours post-transfection cells were stimulated with 2 mM CaCl_2_ modified Ringer’s solution with 2 μM thapsigargin (TG) (Merck, Darmstadt, Germany, Cat#67526-95-8) for 45 min. Cells were fixed with a fixative solution [4% *w*/*v* formaldehyde (freshly prepared from paraformaldehyde, Sigma-Aldrich, San Louis, MO, USA, Cat#158127), 4% w/v sucrose (Sigma-Aldrich, Cat#S0389) in Dulbecco’s phosphate-buffered saline (DPBS), at pH 7.4. After three washes for 10 min each, Hoechst 33342 nuclear stain at 200 ng/mL (Thermo Fisher Scientific, Waltham, MA, USA, Cat#H3570) was performed. Samples were mounted with Fluoromount (Sigma-Aldrich, San Louis, MO, USA, Cat#F4680). Images were acquired with 63× objective 1.35 N.A. using optical sectioning and structured illumination Zeiss Apotome 2, Axiovert 7.

### 4.8. Focal Adhesion Dynamics

Focal adhesion (FAs) turnover was studied via live-cell imaging recordings using total internal reflection fluorescence microscopy (TIRFM) [64,65]. This approach was performed in MDA-MB-231 cells transfected with EGFP-Paxillin and mScarlet or mScarlet-C-SARAF. The cells were transfected in suspension and immediately seeded in fibronectin-coated coverslips (5 μg/mL). The cells were depleted from serum 48 h post-transfection for at least 3 h. Imaging was performed at room temperature. FA turnover dynamics were stimulated by the addition of 10% *v*/*v* FBS in Ringer’s solution modified with 2 mM CaCl_2_. Images were acquired every 1 min for 30 frames using a TIRFM 60× objective (N.A. = 1.45) in an Olympus IX71 microscope equipped with an FLIR camera (Backfly S, BFS-U3-51S5M, FLIR, Richmond, BC, Canada). Images were processed using Image J2/FIJI v.2.9.0 software. FA assembly and disassembly were visualized and quantified as the appearance or loss of fluorescence in a region of interest, according to Goetz et al. (2008) [66]. The assembly and disassembly rates were calculated by plotting the values of ln I/I_0_ versus time as described [65].

### 4.9. Spreading Assay

Spreading assays were performed as described [65]. Briefly, cells were serum-starved 16 h prior to assay. Then, 2 × 10^4^ cells were seeded in 5 μg/mL fibronectin-coated 12 mm coverslips using a medium containing 10% *v*/*v* FBS. Ninety minutes post-seeding, cells were fixed in a fixative solution (4% *w*/*v* formaldehyde—freshly prepared from paraformaldehyde—4% *w*/*v* sucrose in DPBS, pH 7.4) for 15 min at 4 °C. Then, cells were permeabilized and blocked with blocking solution (4% *w*/*v* non-fat dry milk/0.1% *v*/*v* Triton X-100 in DPBS) for 45 min at room temperature. Actin cytoskeleton was detected with Alexa-488-conjugated phalloidin (Thermo Fisher Scientific, Waltham, MA, USA, Cat#A12379). Nuclei were detected staining with Hoechst 33342 reagent. Images were acquired with 63× objective 1.35 N.A. using optical sectioning and structured illumination Zeiss Apotome 2, Axiovert 7 (Carl Zeiss, Oberkochen, Germany).

### 4.10. Proliferation MTT Assay

MDA-MB-231 (3 × 10^3^ cells) and 4T1 cells (2.5 × 10^3^ cells) were seeded in 96-well plates. A total of 0.5 mg/mL MTT reagent (Sigma-Aldrich, San Louis, MO, USA, Cat#2128), was added and then the microplate was incubated for 2 h at 37 °C. Formazan crystals were solubilized with dimethyl sulfoxide. Absorbance was measured at 570 nm (Elx-800, Biotek Instruments Inc., Winooski, VT, USA).

### 4.11. Transwell Chamber Invasion Assays

MDA-MB-231 and 4T1 cells (5 × 10^4^ cells) transfected with mScarlet or mScarlet-C-SARAF were plated into 8 µm pore Transwell^®^ chambers (Sigma-Aldrich, San Louis, MO, USA, Cat#CLS3422). For the invasion assay, Transwell^®^ chambers were preincubated with Matrigel (Sigma-Aldrich, San Louis, MO, USA, Cat#E1270) overnight at 4 °C and then they were placed for 30 min at 37 °C before seeding the cells. Cell invasion was induced by adding 10% *v*/*v* FBS in the lower chamber for 16 h at 37 °C. The non-invasive cells were removed, and the invading cells were fixed and stained with 0.2% *w*/*v* crystal violet dissolved in 10% *v*/*v* methanol. The invading cells were counted and expressed as percentage of control. 

### 4.12. Scratch Assay

MDA-MB-231 and 4T1 cells transfected with mScarlet or mScarlet-C-SARAF were seeded on coverslips pre-coated with 5 µg/mL of Human plasma Fibronectin (Sigma-Aldrich, San Louis, MO, USA, Cat#FC010), forming a monolayer. After 24 h, cells were depleted of serum for 4 h and then scratches were made with a sterile micropipette tip (p200). Detached cells were removed by washing three times with DPBS, and then the medium was replaced with DMEM 2% FBS and three images were obtained from each wound (t_0_) with an inverted microscope. After 16 h, cells were fixed for 15 min at 4 °C in fixative solution [4% *w*/*v* formaldehyde (freshly prepared from paraformaldehyde, Sigma-Aldrich, San Louis, MO, USA, Cat#158127), 4% *w*/*v* sucrose (Sigma-Aldrich, San Louis, MO, USA, Cat#S0389) in Dulbecco’s phosphate-buffered saline (DPBS), pH 7.4], and three images were obtained from each wound (t_16h_). The relative sizes of the wounds at t_0_ and t_16h_ were obtained using a wound-healing size plugin of the Image J2/FIJI v.2.9.0 software and results were expressed as percentage of closing ([(Area t_0h_ − Area t_16h_)/ Area t_0h_] × 100).

### 4.13. In Vivo Metastasis Model

Stable cell lines of 4T1 that overexpress mScarlet or mScarlet-C-SARAF were injected to generate two animal models associated with the animal protocol 20382-MED-UCH, which was approved by the Bioethics Committee of the Faculty of Medicine, Universidad de Chile. Female Balb/c mice were orthotopically injected into the fatty tissue of the fourth mammary gland (inguinal area) with stable cell line 4T1. Cells were resuspended in Dulbecco’s modified phosphate-buffered saline (DPBS) with 0.04% trypan blue (2 × 10^5^ cells, 50 µL) [67]. The general condition of the animals was evaluated daily according to the modified monitoring protocols of Morton and Griffiths (Veterinary Record, 116: 431-36, 1985). The mice were maintained for 21 days or when the score was equal to or greater than 15. Tumor growth was assessed postmortem. Once the tumor was extracted, three diameters of the ellipsoid were measured using a caliper and the following formula was used to calculate the volume [68]:$$V_{ellipsoid}=\frac{4}{3}\;\pi \cdot \frac{x}{2}\cdot \frac{y}{2}\cdot \frac{z}{2}$$

Additionally, organs were removed, fixed, and processed for histological analysis and the evaluation of metastatic sites.

### 4.14. Histology

The main organs that were affected during cancer metastasis, either by indirect effect of primary tumor growth, were removed, washed with DPBS and fixed in fixative solution (4% *w*/*v* formaldehyde—freshly prepared from paraformaldehyde—4% *w*/*v* sucrose in DPBS, pH 7.4) for 24 h at 4 °C. Subsequently, the tissue was processed with increasing concentrations of alcohol, cleared with xylol and embedded in paraffin. Serial sections with a thickness of 3 μm were obtained and adhered to slides for histological hematoxylin and eosin and Masson’s trichrome staining. Images were acquired with a Leica DFC425C camera installed on a Leica DM2500 CCD microscope with 10× and 40× objectives (Leica, Wetzlar, Germany).

### 4.15. Statistical Analyses

Data shown correspond to the mean ± SD or mean ± SEM of at least three independent experiments. Data were analyzed using two-tailed Mann–Whitney’s or Welch’s test to compare two conditions. For multiple comparisons, one-way analysis of variance (ANOVA) tests and Dunnett’s post-test were applied. Analyses were carried out using GraphPad Prism v8.0 (GraphPad Prism, San Diego, CA, USA).

## Figures and Tables

**Figure 1 ijms-24-05306-f001:**
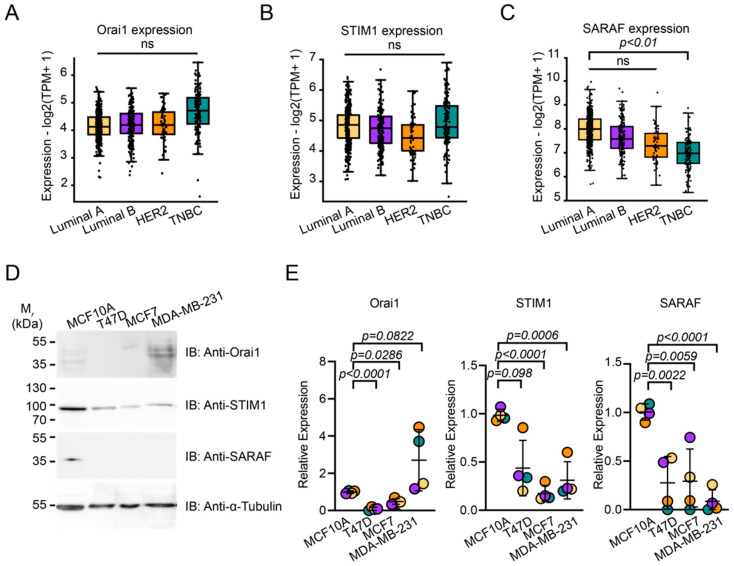
SARAF expression is diminished in triple-negative breast cancer. (**A**–**C**) GEPIA analyses from TCGA database. Relative transcript expression boxplot with jitter graphs of Orai1 (**A**), STIM1 (**B**) and SARAF (**C**) in different subtypes of breast cancer tumors. Each point represents individual human samples. Whiskers represent minimum (bottom) and maximum (upper) values. Number of human samples in luminal A (N = 415), luminal B (N = 194), HER2 (N = 66), and triple-negative cancer samples (N = 134). Statistical analyses correspond to ANOVA according to the GEPIA2 platform (available at http://gepia2.cancer-pku.cn. Accessed on 26 June 2022). (**D**) Immunoblot of protein expression of Orai1, STIM1 and SARAF in breast-related cell lines. (**E**) Relative expression of Orai1, STIM1 and SARAF related to α-tubulin levels (N = 4). Bars represent average expression levels. Error bars represent standard deviation. Individual points represent each experiment. Statistical analyses correspond to ANOVA followed by Dunnett’s multiple comparison test.

**Figure 2 ijms-24-05306-f002:**
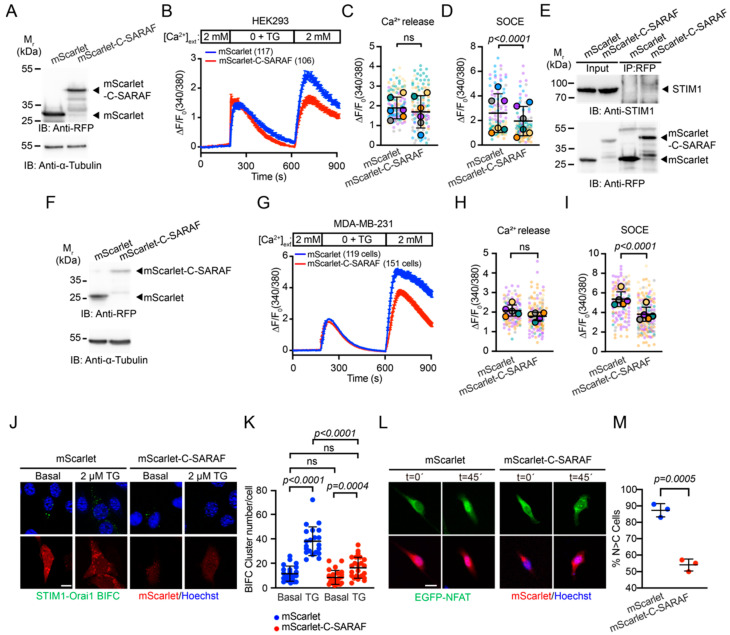
The C-terminal fragment of SARAF reduces SOCE response. (**A**) Representative immunoblot of mScarlet and mScarlet-C-SARAF overexpression in HEK293 cells. (**B**) Mean traces of SOCE response in HEK293 cells overexpressing mScarlet or mScarlet-C-SARAF. Error bars correspond to standard error. (**C**) Thapsigargin-induced Ca^2+^ peak quantification. (**D**) Ca^2+^ influx observed in the SOCE response. Small points represent individual cells. Color circles represent average of each independent experiment. Bar represents the average of individual points. Error bars correspond to standard deviation. Mann–Whitney’s non-parametric *t*-test was applied to define statistical significance. (**E**) Co-immunoprecipitation of mScarlet and mScarlet-C-SARAF and endogenous STIM1 from HEK293 cells in resting conditions. Immunoblots of input and immunoprecipitation (IP) products are shown. Arrowheads indicate STIM1 and mScarlet or mScarlet-C-SARAF bands, respectively. A representative immunoblot of three independent experiments is shown. (**F**) Representative immunoblot of mScarlet and mScarlet-C-SARAF overexpression in MDA-MB-231 cells. (**G**) Representative graph of SOCE response in MDA-MB-231 cells overexpressing mScarlet or mScarlet-C-SARAF. Error bars correspond to standard error. (**H**) Ca^2+^ peak quantification induced by reticular Ca^2+^ depletion in (**G**). (**I**) Quantification of Ca^2+^ influx observed in the SOCE response. Small points represent individual cells. Colored circles represent the average of each independent experiment. Bars represent the average of individual points. Error bars correspond to standard deviation. Statistical analyses correspond to Mann–Whitney’s test. (**J**) Representative panel of the effect of mScarlet or mScarlet-C-SARAF overexpression in MDA-MB-231 cells on BiFC STIM1-Orai1 formation. The cells were treated with 2 µM TG for 45 min at 37 °C in Ringer’s solution modified with 2 mM CaCl_2_ (**K**) Quantification of fluorescence puncta from BiFC STIM1-Orai1 formation. Bars represent average response. Error bars correspond to standard deviation. Scale bar = 20 μm. Individual points represent the response of each cell. Statistical analyses correspond to ANOVA followed by Dunnett’s multiple comparison test. (**L**) Representative panel of the effect of mScarlet or mScarlet-C-SARAF overexpression in MDA-MB-231 cells on EGFP-NFAT translocation. (**M**) Quantification of the translocation of NFAT to the nucleus observed in (**L**). Bars represent average response. Error bars correspond to standard deviation. Scale bar = 20 μm. Individual points represent the average response of each experiment. Statistical analyses correspond to Welch’s test.

**Figure 3 ijms-24-05306-f003:**
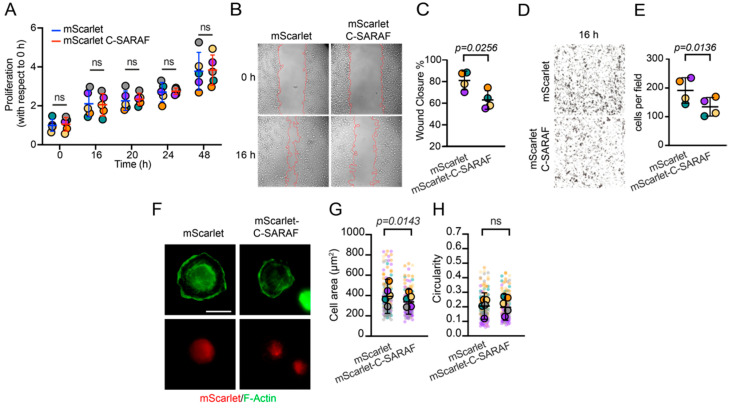
The C-terminal fragment of SARAF reduces malignant features in TNBC. (**A**) MTT proliferation assay of MDA-MB-231 cells overexpressing mScarlet or mScarlet-C-SARAF (N = 5). Statistical analyses correspond to Mann–Whitney’s non-parametric *t*-test. Bars represent the average of individual points. Error bars correspond to standard deviation. (**B**) Representative images of migration assay in MDA-MB-231 overexpressing mScarlet or mScarlet-C-SARAF. (**C**) Quantification of the gap closure (N = 4) observed in (**B**). Bars represent the average of individual points. Error bars correspond to standard deviation. Statistical analysis corresponds to Mann–Whitney’s non-parametric *t*-test. (**D**) Representative images of Transwell^®^ cell invasion assay on cells overexpressing mScarlet or mScarlet-C-SARAF in MDA-MB-231 cells. (**E**) Quantification of invading cells in (**D**). Bars represent the average of individual points. Error bars correspond to standard deviation (N = 4). Statistical analysis corresponds to Mann–Whitney’s non-parametric *t*-test, which was applied to define statistical significance. (**F**) Representative panels of the effect of mScarlet or mScarlet-C-SARAF overexpression on the cell spreading of MDA-MB-231 cells (N = 5). (**G**) Graph of total cell area of MDA-MB-231 cells. (**H**) Graph indicating circularity of MDA-MB-231 cells. Mann–Whitney’s non-parametric *t*-test was applied to define statistical significance. Bars represent average response. Individual points represent the response of each cell. Colored circles represent the average of each independent experiment. Scale bar = 20 μm.

**Figure 4 ijms-24-05306-f004:**
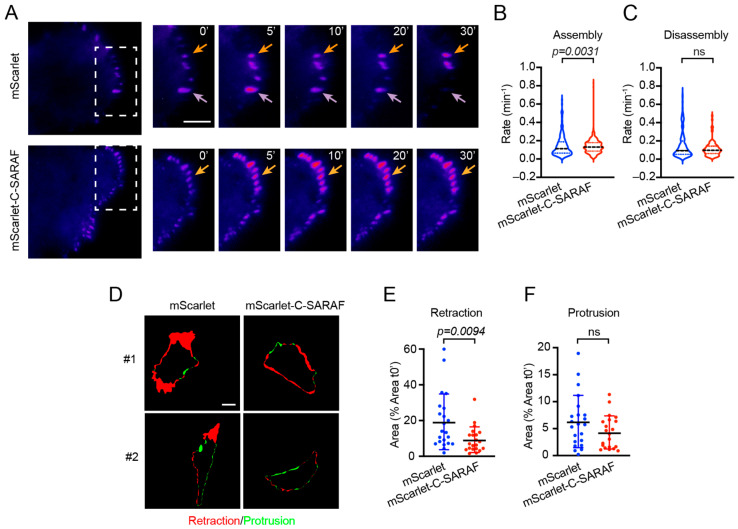
The C-terminal fragment of SARAF increases the rate of assembly of focal adhesions and reduces cell retraction. (**A**) Representative time-lapse imaging of EGFP-Paxillin from MDA-MB-231 cells co-expressing mScarlet or mScarlet-C-SARAF. Assembling FAs are marked with orange arrows, while purple arrows mark assembling and disassembling focal adhesions. Scale bar = 5 μm. Quantification of assembly (**B**) and disassembly rates (**C**) of FAs from recordings of EGFP-Paxillin transfected MDA-MB-231. (**D**) Representative images from two cells per condition of retraction (red) and protrusion (green) of MDA-MB-231 cells observed in (**A**). Quantification of retraction (**E**) and protrusion (**F**) cell area. Scale bar = 20 μm. Bars represent average response. Individual points represent the response of each cell. Bar represents the average of individual points. Error bars correspond to standard deviation. Mann–Whitney’s non-parametric *t*-test was applied to define statistical significance.

**Figure 5 ijms-24-05306-f005:**
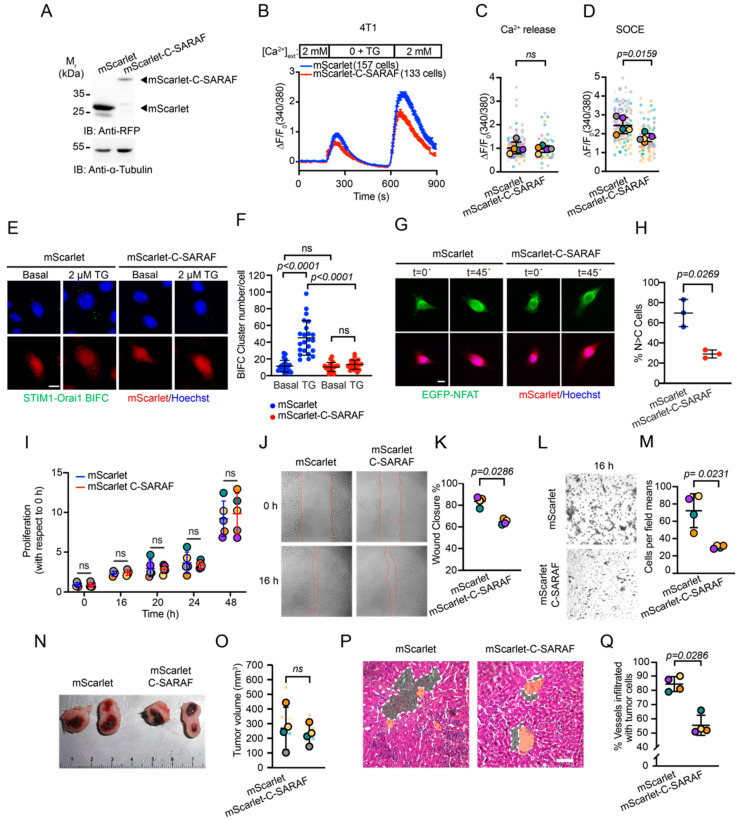
The C-terminal fragment of SARAF reduces SOCE response and affects metastasis and tumor-infiltrating lymphocyte in tumoral growth in vivo murine model. (**A**) Representative immunoblot of mScarlet and mScarlet-C-SARAF overexpression in 4T1 cells. (**B**) Mean traces of SOCE response in 4T1 cells overexpressing mScarlet or mScarlet-C-SARAF. (**C**) Ca^2+^ signal peak quantification induced by ER depletion in (**B**). (**D**) Quantification of Ca^2+^ influx observed in the SOCE response (**B**). Mann–Whitney’s non-parametric *t*-test was applied to define statistical significance. (**E**) Representative panel of the effect of mScarlet or mScarlet-C-SARAF overexpression in 4T1 cells on BiFC STIM1-Orai1 formation. The cells were treated with 2 µM TG for 45 min at 37 °C in Ringer’s solution modified with 2 mM CaCl_2_ (**F**) Quantification of fluorescence puncta from BiFC STIM1-Orai1 formation (N = 4). ANOVA and Dunnett’s multiple comparison test were applied to define statistical significance. Scale bar = 20 μm. (**G**) Representative images of the effect of mScarlet or mScarlet-C-SARAF overexpression in MDA-MB-231 cells on EGFP-NFAT translocation. (**H**) Quantification of the translocation of NFAT to the nucleus observed in (**G**) (N=3). Bars represent average response. Error bars correspond to standard deviation. Scale bar = 20 μm. Individual points represent the average response of each experiment. Colored circles represent the average of each independent experiment. Statistical analyses correspond to Welch’s test. (**I**) MTT proliferation assay of 4T1 cells overexpressing mScarlet or mScarlet-C-SARAF (N = 5). Mann–Whitney’s non-parametric *t*-test was applied to define statistical significance. (**J**) Representative images of migration assay in 4T1 cells overexpressing mScarlet or mScarlet-C-SARAF. (**K**) Quantification of the gap closure (N = 4) observed in (**J**). Mann–Whitney’s non-parametric *t*-test was applied to define statistical significance. (**L**) Representative images of Transwell^®^ cell invasion assay on cells overexpressing mScarlet or mScarlet-C-SARAF in 4T1 cells. (M) Quantification of invading cells in (**L**) (N = 4). Mann–Whitney’s non-parametric *t*-test was applied to define statistical significance. (**M**) Representative image of primary tumors obtained after injection of stable lines of 4T1 overexpressing mScarlet or mScarlet-C-SARAF in the fourth mammary gland of Balb/c mice. (**N**) Graph of primary tumor volume (N = 4). Statistical analysis where Mann–Whitney non-parametric *t*-test was applied to define statistical significance. (**O**) Histological section from liver samples stained with hematoxylin and eosin. Green region corresponds to tumoral cell infiltration, and the orange region is vessel lumen. (**P**) Quantification of percentage of portal vessels infiltrated with tumoral cells observed in (**Q**). Mann–Whitney’s non-parametric *t*-test was applied to define statistical significance.

**Table 1 ijms-24-05306-t001:** Plasmids.

Plasmid	Source	Catalog #	Amino Acid Sequence Information
SARAF-myc-DDK	Origene	RC201864	-
mScarlet-H	Addgene	85043	-
mScarlet-C-SARAF	Cerda Lab	-	cytosolic domain SARAF (195–339 aa) fused to mScarlet protein
EGFP-Paxillin	Dr. Christopher Turner		
EGFPC1-huNFATc1EE-WT	Addgene	24219	-
pcDNA3-Venus-173-N-Orai1	Addgene	87618	-
pcDNA3.1-STIM1-Venus-173-C	Addgene	87619	-

**Table 2 ijms-24-05306-t002:** Antibodies.

Antibody	Isotype	Dilution	Final [], µg/mL	Source	Type	Catalog	RRID	Purification
Mouse anti-Orai1	IgG1	1:1000 (IB)	1	Santa Cruz Biotechnology	mAb	sc-377281	AB_2797381	-
Mouse Anti-STIM1	IgG2a	1:1000 (IB)		Origene	mAb	AM20946PU-N	AB_10849514	-
Mouse anti-α-tubulin	IgG1	1:5000 (IB)	1	Sigma Aldrich	mAb	T5168	AB_477579	Asc
Mouse Anti-RFP [6G6]	IgG2c	1:2000 (IB) 1:100 (IP)		ChromoTek	mAb	6g6	AB_2631395	AP
Rat Anti-RFP [5F8]	IgG2a	1:100 (IP)		ChromoTek	mAb	5f8	AB_2336064	AP
Rabbit Anti-STIM	IgG	1:2000 (IB)		Cell Signaling	mAb	D88E10	AB_10828699	
Rabbit Anti-SARAF (TMEM66)	IgG	1:250 (IB)	–	Origene	pAb	TA341997	–	AP
Goat anti-Rabbit HRP-conjugated Antibody	IgG	1:5000 (IB)	–	R&D systems	pAb	HAF008	AB_357235	AP
Goat anti-mouse HRP-conjugated Antibody	IgG	1:5000 (IB)	–	R&D systems	pAb	HAF007	AB_357234	AP

Abbreviations: AP, affinity purified; Asc, purified from mouse ascites fluids via affinity chromatography; IB, immunoblot; IP, immunoprecipitation; mAb, monoclonal antibody; pAb, polyclonal antibody.

## Data Availability

Not applicable.

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
