# Peer review of "The Cytoplasmic Region of SARAF Reduces Triple-Negative Breast Cancer Metastasis through the Regulation of Store-Operated Calcium Entry"

_ijms, 2023, doi:10.3390/ijms24065306_

Round 1
Reviewer 1 Report
The present manuscript reported the role of C-terminal region of SARAF as a negative regulator of SOCE in breast cancer and the subsequent reduction in cancer hallmarks development, including cell migration and invasion. The study is very interesting, well organized and conducted. The authors suggested the relevance of the knowledge arised from their results and its future application to identify new therapies against for triple-negative breast cancer. Nonetheless, it is important that authors address the concerns indicated before considering its publication in the journal.
Mayor Comments:
1. Authors demonstrate the involvement of C-terminal region of SARAF in STIM1-Orai1 interaction by overexpressing mScarlet-C-SARAF and using a biomolecular fluorescence complementation (BiFC) assay in which pcDNA3.1-STIM1-Venus-173-C and pcDNA3-Venus-173-N-Orai1 are also overexpressed. It would be interesting to demonstrate this finding by immunoprecipitation assay in cells transfected only with mScarlet-C-SARAF and to analyze the interaction of native proteins (Orai1-STIM1).
2. In the Discussion (Page 10; line 350), authors indicates that “the mScarlet-C-SARAF fragments could interact directly with STIM1, regulating its conformational change, which may result in reduced cell migration and invasion of cancer cells”. To demonstrate this mechanism, author should demonstrate this interaction.
Minor Comments:
1. Page 3, Line 110. The Sentence “As the MCF10A line is a non-neoplastic cell line, we observed a decrease in SARAF and STIM1 expression in MCF7 and MDA-MB-231 cells” should be rewritten since, it is shown in figure 1E, there are no significant differences in the STIM1 expression in MDA-MB-231 cells compared to MCF10A. Despite the difference not being significant, the trend observed in the figure is the decrease of STIM1 expression in tumoral cells compared with non-tumoral cells. It would be recommendable to increase the number of experiments performed to obtain significant differences in the expression of STIM1 between both cell types.
2. Page 4, line 141. “thapsigargin-induced Ca2+ signaling” should be change by “thapsigargin-induced Ca2+ release”
3. In Figure 2C and 2G, “TG” should be change by “Ca2+ release”.
4. Page 5, line 168. “(E)” should be deleted.
5. In figure 5C “TG-induced” should be change by “Ca2+ release”.
6. Page 9 line 329 the sentence “A reduction in the expression of STIM1 and Orai1 reduces the migration and invasion of cells, and it causes a reduction of metastasis in a murine model, as suggested by previous studies” should be rewritten. Redundant language (reduction/reduces/reduction)
Author Response
Reviewer Comments:
Reviewer #1:
The present manuscript reported the role of C-terminal region of SARAF as a negative regulator of SOCE in breast cancer and the subsequent reduction in cancer hallmarks development, including cell migration and invasion. The study is very interesting, well organized and conducted. The authors suggested the relevance of the knowledge arised from their results and its future application to identify new therapies against for triple-negative breast cancer. Nonetheless, it is important that authors address the concerns indicated before considering its publication in the journal.
R: Thank you for your careful revision. We have addressed all the reviewer’s points, as detailed:
Major Comments:
- Authors demonstrate the involvement of C-terminal region of SARAF in STIM1-Orai1 interaction by overexpressing mScarlet-C-SARAF and using a biomolecular fluorescence complementation (BiFC) assay in which pcDNA3.1-STIM1-Venus-173-C and pcDNA3-Venus-173-N-Orai1 are also overexpressed. It would be interesting to demonstrate this finding by immunoprecipitation assay in cells transfected only with mScarlet-C-SARAF and to analyze the interaction of native proteins (Orai1-STIM1).
R: We thank the reviewer’s suggestion. During the revision, we attempted to perform these experiments. Unfortunately, the antibodies were not suitable for immunoprecipitation experiments. We could try these experiments with other antibodies in our future research. Despite that, the functional experiments with the native SOCE system in the three cell lines assayed support the effect. Also, these experiments are consistent with the Orai1-STIM1 BiFC and NFAT translocation assays, demonstrating the effect of the mScarlet-C-SARAF on SOCE inhibition.
- In the Discussion (Page 10; line 350), authors indicates that “the mScarlet-C-SARAF fragments could interact directly with STIM1, regulating its conformational change, which may result in reduced cell migration and invasion of cancer cells”. To demonstrate this mechanism, author should demonstrate this interaction.
R: We appreciate the reviewer’s commentary. We performed and included these experiments (please see Figure 2E, page 4, lines 8-9)
Minor Comments:
- Page 3, Line 110. The Sentence “As the MCF10A line is a non-neoplastic cell line, we observed a decrease in SARAF and STIM1 expression in MCF7 and MDA-MB-231 cells” should be rewritten since, it is shown in figure 1E, there are no significant differences in the STIM1 expression in MDA-MB-231 cells compared to MCF10A. Despite the difference not being significant, the trend observed in the figure is the decrease of STIM1 expression in tumoral cells compared with non-tumoral cells. It would be recommendable to increase the number of experiments performed to obtain significant differences in the expression of STIM1 between both cell types.
R: Thank you for the suggestion. We have repeated these experiments and corrected the figure accordingly (please see Figures 1D-E, page 3, lines 31-33).
- Page 4, line 141. “thapsigargin-induced Ca2+ signaling” should be change by “thapsigargin-induced Ca2+ release”
R: Thank you for your commentary. We have corrected the figure accordingly.
- In Figure 2C and 2G, “TG” should be change by “Ca2+ release”.
R: Thank you for your commentary. We have corrected the figure accordingly.
- Page 5, line 168. “(E)” should be deleted.
R: Thank you for the suggestion. We corrected the text according the suggestion.
- In figure 5C “TG-induced” should be change by “Ca2+ release”.
R: Thank you for your commentary. We have corrected the figure accordingly.
- Page 9 line 329 the sentence “A reduction in the expression of STIM1 and Orai1 reduces the migration and invasion of cells, and it causes a reduction of metastasis in a murine model, as suggested by previous studies” should be rewritten. Redundant language (reduction/reduces/reduction).
R: Thank you for your commentary. We have corrected the sentence according the reviewer’s suggestion. Please refer to Page 10, lines 1-2)
Reviewer 2 Report
María Paz Saldías and co-authors have evaluated the effect of the over-expression of C-term fragment of SARAF on SOCE as well as proliferation, migration and invasion of MDA-MB-231 TNBC cell line and 4T1 murine mammary carcinoma cell line. The claims made in the manuscript are supported by the data presented. I do not have any hesitations in recommending the study for a publication. Could the authors please comment on if they have considered to test the effect of SARAF C-term over-expression or SARAF knock-down in MCF10A cells on SOCE as well as associated properties that were tested for the TNBCs?
Author Response
Reviewer #2:
María Paz Saldías and co-authors have evaluated the effect of the over-expression of C-term fragment of SARAF on SOCE as well as proliferation, migration and invasion of MDA-MB-231 TNBC cell line and 4T1 murine mammary carcinoma cell line. The claims made in the manuscript are supported by the data presented. I do not have any hesitations in recommending the study for a publication. Could the authors please comment on if they have considered to test the effect of SARAF C-term over-expression or SARAF knock-down in MCF10A cells on SOCE as well as associated properties that were tested for the TNBCs?
R: We thank to the reviewer for the positive commentaries. We appreciate the reviewer’s commentary. It would be interesting to test the effect of SARAF silencing and/or SARAF C-term expression. Accordingly, we would expect that SARAF silencing could increase SOCE response, and invasive behavior in MCF10A cells. We could explore these effects on our future research.
Round 2
Reviewer 1 Report
I thank the authors for taking my comments about the manuscript into account, as well as the effort made to answer them. However, before accepting the article for publication, I have two minor comments about Figure 2.
1. Figure 2E. Authors should indicate in the main text and/or figure legend that experiments were performed in resting conditions ?
2. Figure 2J. Authors should indicate in the main text and/or figure legend the incubation time with thapsigargin.
Author Response
Reviewer #1:
I thank the authors for taking my comments about the manuscript into account, as well as the effort made to answer them. However, before accepting the article for publication, I have two minor comments about Figure 2.
R: Thank you for your careful revision. We have addressed all the reviewer’s points, as detailed:
- Figure 2E. Authors should indicate in the main text and/or figure legend that experiments were performed in resting conditions ?
R: Thank you for your commentary. We have modified the legend accordingly.
- Figure 2J. Authors should indicate in the main text and/or figure legend the incubation time with thapsigargin.
R: Thank you for your commentary. The incubation time and conditions are included in the Material and Methods section. However, we have included this information in legends for Fig 2J and 5E.